# Combined Modeling of the Optical Anisotropy of Porous Thin Films

**F. V. Grigoriev** [1,2,*]**, V. B. Sulimov** [1,2] **and A.V. Tikhonravov** [1,2]

[1]   Research Computing Center, M. V. Lomonosov Moscow State University, Leninskie Gory, 1, 119991 Moscow, Russia; vs@dimonta.com (V.B.S.); tikh@srcc.msu.ru (A.V.T.)

[2]   Moscow Center of Fundamental and Applied Mathematics, Leninskie Gory, 1, 119234 Moscow, Russia

*   Correspondence: fedor.grigoriev@gmail.com

**Abstract:** In this article, a combined approach for studying the optical anisotropy of porous thin films obtained by the glancing angle deposition is presented. This approach combines modeling on the atomistic and continuum levels. First, thin films clusters are obtained using the full-atomistic molecular dynamics simulation of the deposition process. Then, these clusters are represented as a medium with anisotropic pores, the shapes parameters of which are determined using the Monte Carlo based method. The difference in the main components of the refractive index is calculated in the framework of the anisotropic Bruggeman effective medium theory. The presented approach is tested and validated by comparing the analytical and simulation results for the model problems, and then is applied to silicon dioxide thin films. It is found that the maximum difference between the main components of the refractive index is 0.035 in a film deposited at an angle of 80°. The simulation results agree with the experimental data reported in the literature.

**Keywords:** anisotropy; thin films; glancing angle deposition; refractive index; molecular dynamics

## 1. Introduction

The deposition in a vacuum is the most widely used technique for thin film production [1]. The structural and optical properties of the deposited films depend on the deposition condition, in particular, on the angle between the direction of atoms flow and normal to the substrate surface [2]. Deposition at large angles, when atoms move almost parallel to the substrate, leads to the formation of glancing angle deposited (GLAD) films consisting of the separate nanostructures with different shapes and dimensions [3]. Due to the low reflectance and anisotropy of the refractive index, GLAD films are widely used in optical instruments [4–8].

To describe the anisotropic properties of the deposited films, the anisotropic Bruggeman effective medium approach is usually used [6,9–14]. The Bruggeman formalism was initially developed for the calculation of the effective dielectric constant of a medium with randomly distributed spherical inclusions [15]. The values of the dielectric constant of medium and inclusions are different. In Reference [16], the approach was generalized for the case of the ellipsoidal inclusions. The anisotropy of a medium with inclusions was taken into account through depolarization factors depending on the ellipsoids shape parameters.

The voids between separated nanostructures in GLAD films can be considered as inclusions of different shapes and dimensions in the films' matter. Within the framework of the Bruggeman's formalism, these inclusions are represented by a set of identical ellipsoids oriented in the same direction. Since the experimental determination of the shape parameters of ellipsoids is still a challenge, these parameters are used to fit experimental results within effective medium models [6,10–14]. On the other hand, shape parameters can be calculated through the Cartezian coordinates of atoms in the clusters

obtained by the simulation of the thin film deposition process. Currently, the different methods of the atomistic level, including the ballistic approach [17,18], molecular dynamics (MD) [19,20], and kinetic Monte Carlo (kMC) [14] method were applied to study the deposition process, structural and mechanical properties of GLAD films. In Reference [14], shape parameters were defined for the titanium dioxide GLAD films using the structure tensor, calculated by the integral of density gradient [21]. The main components of this tensor correspond to the averaged inverse shape parameters. The planar birefringence, important in GLAD films applications for polarization optic [22,23], was calculated using the shape parameters and was compared with the experimental data from Reference [14].

In the present work, a combined approach for modeling of the optical anisotropy of porous thin films is presented. First, the MD-based procedure developed earlier in References [24–26] is used to simulate the high-energy deposition of silicon dioxide thin films. Further, the averaged pore shape parameters determining the anisotropy properties of porous thin films are calculated using the Monte Carlo based method. The method is quite general and is characterized by high computational efficiency. It can be applied to any porous atomistic clusters, obtained using MD, kMC or ballistic approach and consisting of up to $10^7$–$10^8$ atoms. Finally, the values of the difference between the main components of the refractive index $\Delta n$ are calculated in the framework of the anisotropic Bruggeman effective medium approach. The approach is validated by comparing the analytical and simulation results for model problems. The obtained results agree with experimental data for silicon dioxide thin films.

## 2. Simulation Method

This section presents a combined approach to calculating $\Delta n$. We start from the description of the MD procedure that is used in the simulation of thin film growth. Then a brief overview of the anisotropic Bruggeman effective medium theory as applied to the porous atomistic clusters is given. In the final subsection, an original Monte Carlo based approach to the calculation of the averaged shape parameters is described.

### 2.1. Molecular Dynamics Simulation of Thin Film Deposition Process

The simulation of thin film growth is the most time-consuming step in the $\Delta n$ calculation. Nevertheless, the use of parallel computations allows one to increase the dimensions of clusters in the full-atomistic MD simulation up to several tens of nanometers. An increase in cluster dimensions is especially significant in the case of GLAD films consisting of separate nanoscale structures.

The geometry of the deposition process is shown in Figure 1. The simulation area includes parts of the substrate, film and gas phase above growing film. An increase in the deposition angle $\alpha$ leads to an increase in the porosity of the film and anisotropy of the structure. In particular, deposition at $\alpha = 80°$ leads to the formation of an anisotropic structure with separated slanted columns (Figure 1). For this reason, the dependence of $\Delta n$ on $\alpha$ is investigated in a wide range of $\alpha$ from 0° to 80°. In this work, we use clusters that were simulated in our previous works [25,26], except for the cluster deposited at an angle $\alpha = 50°$.

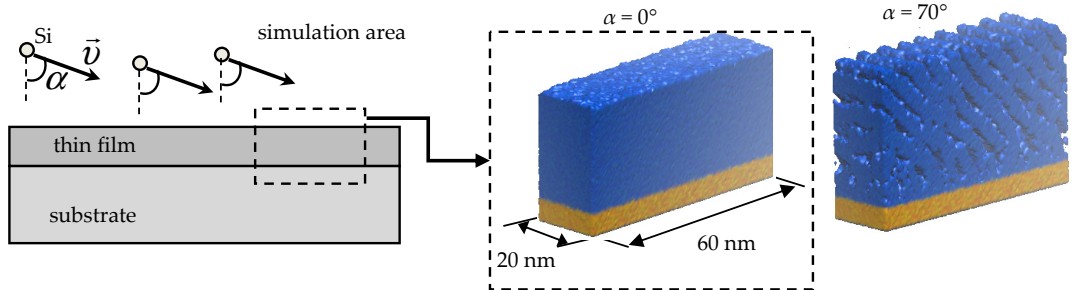

**Figure 1.** To the description of the simulation of thin film growth, $\alpha$ is the deposition angle.

All silicon dioxide film clusters were obtained using the MD-based step by step method, as described in Reference [27]. The energy of interatomic interactions was calculated in the frame of the DESIL force field [27]:

$$U = q_i q_j / r_{ij} + A_{ij} / r_{ij}{}^{12} - B_{ij} / r_{ij}{}^{6} \tag{1}$$

where $q_{i(j)}$ charge of $i(j)$-th atom, $q_O = -0.5q_{Si} = -0.65e$, $A_{ij}$ and $B_{ij}$, parameters of Lennard-Jones potential for the van der Waals interaction, $r_{ij}$ — interatomic distance, $A_{SiO} = 4.6 \times 10^{-8}$ kJ·nm$^{12}$/mol, $A_{SiSi} = A_{OO} = 1.5 \times 10^{-6}$ kJ·nm$^{12}$/mol, $B_{SiO} = 4.2 \times 10^{-3}$ kJ·nm$^6$/mol, $B_{SiSi} = B_{OO} = 5 \times 10^{-5}$ kJ·nm$^6$/mol. MD simulation is performed using the GROMACS package [28]. The leap-frog algorithm is used for the integration of particles motion equations [29].

Horizontal dimensions of the substrate are 20 nm and 60 nm, the vertical thickness of the substrate is taken equal to 6 nm (Figure 1). At each injection step, silicon and oxygen atoms with the stoichiometric proportion of 1:2 are inserted randomly at the top of the simulation box. Then injected atoms move toward the substrate and form new sub-layers on the surface of the growing film. The maximum value of the deposited film thicknesses is about 45 nm, which corresponds to about 4000 injection steps. The initial values of the silicon and oxygen atoms kinetic energies are $E(Si) = 10$ eV and $E(O) = 0.1$ eV, which corresponds to the high-energy deposition processes like ion beam sputtering (IBS) [3]. At each injection step, the *NVT* (constant number of particles, volume and temperature) ensemble is used. The vertical dimension of the simulation box is increased by 0.01 nm after each injection step in order to compensate for the growth of film thickness. The horizontal dimensions of the box remain unchanged. The Berendsen thermostat [30] is applied to keep the simulation cluster temperature, $T = 300$ K, constant. The duration of one injection step is 6 ps, and the time step of MD modeling is 0.5 fs.

The simulation was carried out using the equipment of the shared research facilities of HPC computing resources at Lomonosov Moscow State University [31].

### 2.2. Bruggeman Effective Medium Theory for the Porous Structures

In the framework of the anisotropic Bruggeman effective medium approach, the pores in the structure are approximated by a set of ellipsoids [9]. These ellipsoids are randomly distributed over the film volume and have the same shape parameters $a_x$, $a_y$, $a_z$ and the same orientations (Figure 2). The original method for the calculation of shape parameters is described in Section 2.3.

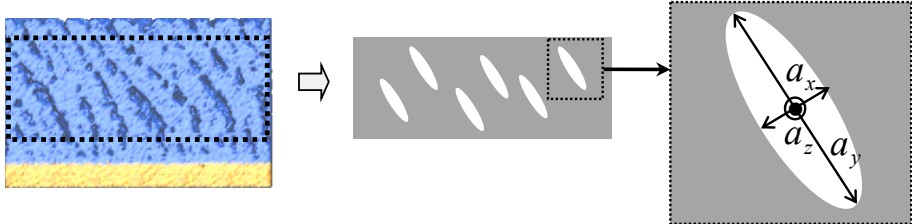

**Figure 2.** Representation of the porous structure in the framework of the anisotropic Bruggeman effective medium approach. The shape parameters of the ellipsoids $a_x$, $a_y$ and $a_z$ are calculated by averaging the pores' dimensions along coordinate system axes.

After determining the values of $a_x$, $a_y$ and $a_z$, the depolarization factors $L_{x(y,z)}$ are calculated as follows [9,16]:

$$L_{x(y,z)} = \frac{a_x a_y a_z}{2} \int_0^\infty \frac{dx'}{\left(x' + a_x^2\left(a_y^2, a_z^2\right)\right)\sqrt{\left(x' + a_x^2\right)\left(x' + a_y^2\right)\left(x' + a_z^2\right)}} \tag{2}$$

where $x'$ is the integration parameter. These factors depend only on the ratios of $a_x$, $a_y$ and $a_z$. Indeed, using the substitution $x' = a_x{}^2 t$, we obtain for $L_x$:

$$L_x = \frac{1}{2} \int_0^{\infty} \frac{dt}{(1+t)\sqrt{(1+t)\left(1 + ta_x^2/a_y^2\right)\left(1 + ta_x^2/a_z^2\right)}} \tag{3}$$

where $t$ is the dimensionless variable. Similar equations can be obtained for others depolarization factors.

The main components of the dielectric constant tensor $\varepsilon_{x(y,z)}$ of the medium consisting of $k$ materials are calculated as follows [9,16]:

$$\sum_{i=1}^{k} f_i \frac{\varepsilon_i - \varepsilon_{x(y,z)}}{\varepsilon_{x(y,z)} + L_{x(y,z)}\left(\varepsilon_i - \varepsilon_{x(y,z)}\right)} = 0 \tag{4}$$

where $f_i = V_i/V$ and $\varepsilon_i$ are the volume fraction and dielectric constant of the $i$-th material. The components of the refractive index $n_{x(y,z)}$ are calculated using the Maxwell relation between refractive index and dielectric constant $n^2 = \varepsilon$.

For the case of porous films Equation (4) is simplified as follows:

$$f_1 \frac{1 - \varepsilon_{x(y,z)}}{\varepsilon_{x(y,z)} + L_{x(y,z)}\left(1 - \varepsilon_{x(y,z)}\right)} + (1 - f_1) \frac{\varepsilon_{df} - \varepsilon_{x(y,z)}}{\varepsilon_{x(y,z)} + L_{x(y,z)}\left(\varepsilon_{df} - \varepsilon_{x(y,z)}\right)} = 0 \tag{5}$$

where $\varepsilon_{df} = 2.22$ [32] is the dielectric constant of dense silicon dioxide film, $f_1$ is the free volume fraction of film. Value of $f_1$ is calculated using the film density dependence on the deposition angle $\rho(\alpha)$:

$$f_1(\alpha) = 1 - \rho(\alpha)/\rho(0) \tag{6}$$

In Equation (6), it is assumed that a normally deposited film ($\alpha = 0$) has no voids and the density of the dense fraction in films, deposited under different $\alpha$, is equal to $\rho(0)$.

### 2.3. Calculation of the Averaged Shape Parameters of Pores in Deposited Film

A sample of the growing film structure is shown in the right part of Figure 3. The elongated pores are formed between columns and are highlighted in dark blue. It is seen that the pores are slanted approximately in the same direction.

In the framework of the anisotropic Bruggeman effective medium approach, it is necessary to calculate the averaged pores dimensions $a_x$, $a_y$ and $a_z$ along the axes of the coordinate system. Furthermore, these parameters are considered as ellipsoids shape parameters in calculating the depolarization factors, see Equations (2) and (3). The coordinate system can be defined by the plane of incidence ($x,z$) and the plane ($x,y$) slanted to the substrate plane at an angle $\beta$ [9]. The $Y$ axis is directed parallel to the substrate plane and perpendicular to the incidence plane, due to the symmetry of the deposition process geometry (Figure 3). The $Z$ axis should be oriented along the direction in which the pores are elongated. This direction is given by the angle $\beta_m$ at which the ratio $a_z/a_x$ reaches its maximum.

- The procedure for calculating the $a_{x(y,z)}$ and $n_{x(y,z)}$ values consists of the following steps:
- The initial value of $\beta$ angle is chosen.
- Set of $N$ points with random coordinates ($x_i$, $y_i$, $z_i$) inside the cluster is generated. For each of the $N_{in}$ points located inside the pores, the pore dimensions $a_{xi}$, $a_{yi}$, and $a_{zi}$ are calculated, as shown in Figure 3. The periodic boundary conditions in the horizontal plane are used (see dotted rectangles in the top right part of Figure 3). In the vertical direction, the dimensions of the open pores are limited by the upper boundary of the film.

- The averaged pores dimensions $a_{x(y,z)}$, are calculated as follows:

$$a_{x(y,z)}(N_{in}) = \frac{\sum_{i=1}^{N_{in}} a_{x(y,z)i}}{N_{in}} \tag{7}$$

When calculating Equation (7), there is no need to determine the volume of every individual pore, which ensures high computational efficiency of the algorithm.

- Step 2 and 3 are repeated with growing $N_{in}$ values. The procedure is finished when $a_{x(y,z)}(N_{in})$ are varying insignificantly with a further increase in $N_{in}$. The ratio $a_z/a_x$ is calculated.
- The values of $L_{x(y,z)}$ and $n_{x(y,z)}$ are calculated using Equations (3) and (4). The difference of the main components of the refractive index $\Delta n$ is calculated.
- The value of β is changed, and steps 2–5 are repeated.
- The $a_{x(y,z)}(N_{in})$ values calculated using β, corresponding to the maximum of $a_z/a_x$ ratio, are the shape parameters.

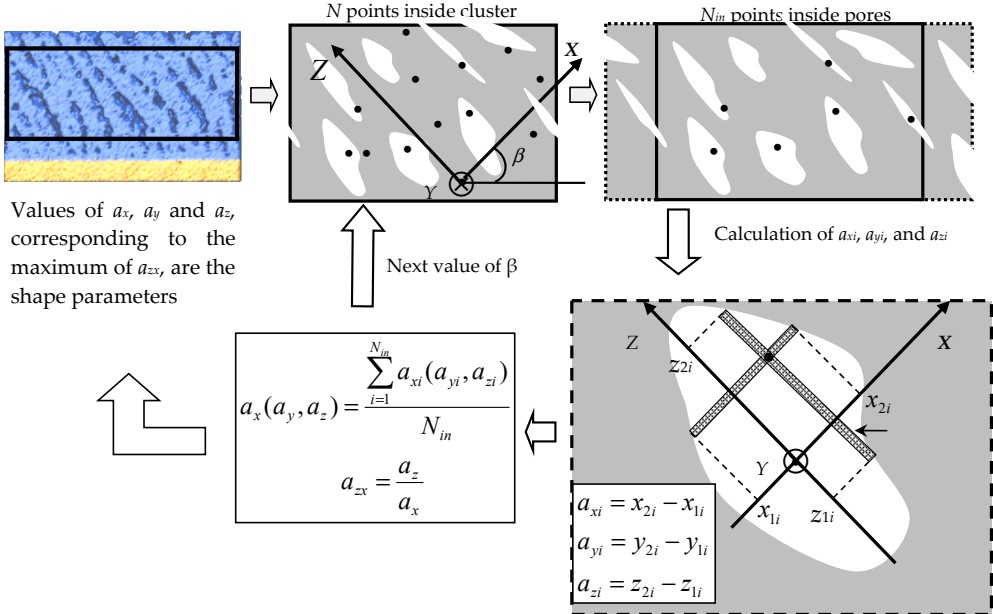

**Figure 3.** The flowchart for calculating the shape parameters $a_x$, $a_y$ and $a_z$ by Monte Carlo averaging of pore dimensions $a_{xi}$, $a_{yi}$ and $a_{zi}$ by volume. The β angle is chosen so as to maximize the $a_z/a_x$ ratio.

Let us prove that Equation (7) gives the $a_{x(y,z)}$ values averaged over the pores volume. Indeed, the volume element $dV(a_z)$, including all of the points giving the same value of $a_z$, is defined in Figure 3:

$$dV(a_z) = a_z(x,y)dxdy \tag{8}$$

In the frame of the Monte Carlo method [33] and in the $N_{in} \to \infty$ limit, the next condition is satisfied:

$$\frac{N(a_z)}{N_{in}} = \frac{dV(a_z)}{V_p} \tag{9}$$

where $N(a_z)$ is the number of points, giving $a_z$ value. Thus, the summation in (7) can be replaced by integration:

$$a_z = \lim_{N_{in}\to\infty} \frac{\sum_{i=1}^{N_{in}} a_{zi}}{N_{in}} = \frac{\iint a_z(x,y)dV(a_z)}{V_p} = \frac{\iint a_z^2(x,y)dxdy}{V_p} \tag{10}$$

The values of $a_{y(z)}$ are calculated in a similar way. For the case of ellipsoidal pores of different dimensions, oriented in the same direction, Equation (10) gives values of $a_{x(y,z)}$ which are one and a half times higher than the averaged values of the ellipsoids semi-axes (Appendix A). However, this difference does not matter, since only ratios of $a_x$, $a_y$ and $a_z$ are required in Equation (3).

The pore boundary search algorithm takes into account that every atom in the clusters is considered as a hard sphere with a certain radius $R$. The values of the atomic radii are discussed in the text above Table 1, in Section 3. Thus, the pore surface consists of fragments of spherical surfaces centered on the atoms. To calculate the $a_{x(y,z)I}$ values that are required in Equation (7), the next algorithm is used:

- The coordinate system is centered in the probe point, which is randomly chosen.

- The distance from each atom to the $X$ axis is equal to $r_A = \sqrt{z_A^2 + y_A^2}$.

- If $r_A$ is less than atomic radius $R$, the distance from the probe point to the point of intersection of the $X$ axis with the sphere of radius $R$, centered on the atom, is calculated as follows (Figure 4):

$$x_{2A} = x_A - \sqrt{R^2 - r_A^2} \tag{11}$$

- The minimum value of $x_{2A}$ is equal to $x_{2i}$.
- The $x_{1i}$ value is calculated similarly. Then the $a_{xi}$ value is calculated as $a_{xi} = x_{2i} - x_{1i}$.
- The $y_{1(2)i}$ and $z_{1(2)i}$ values are calculated in a similar way. The values of $a_x$, $a_y$ and $a_z$ are calculated according to Equation (7).

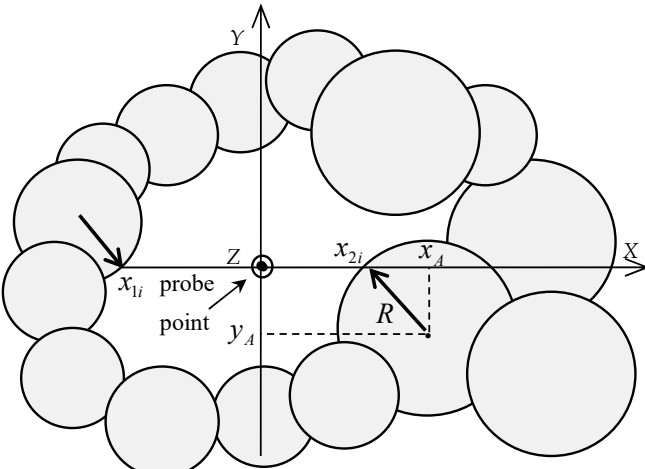

**Figure 4.** Calculation of the pore dimension $a_{xi}$ along $X$ axis in the atomistic cluster. Spheres are centered on the atoms. The sphere radius depends on the chemical element to which the atom belongs.

## 3. Results and Discussion

To validate the method for calculating shape parameters, two model geometries of the pores are considered: A spherical pore and a set of ellipsoidal pores of various dimensions (right side of Figure 5). Pores are created in the atomistic cluster of silicon dioxide film obtained by high-energy deposition. Horizontal dimensions of the cluster are 20 nm and 60 nm, the vertical dimension is 30 nm, and the directions of the coordinate axes can be seen in Figure 3. Atoms whose centers are inside the pores are removed.

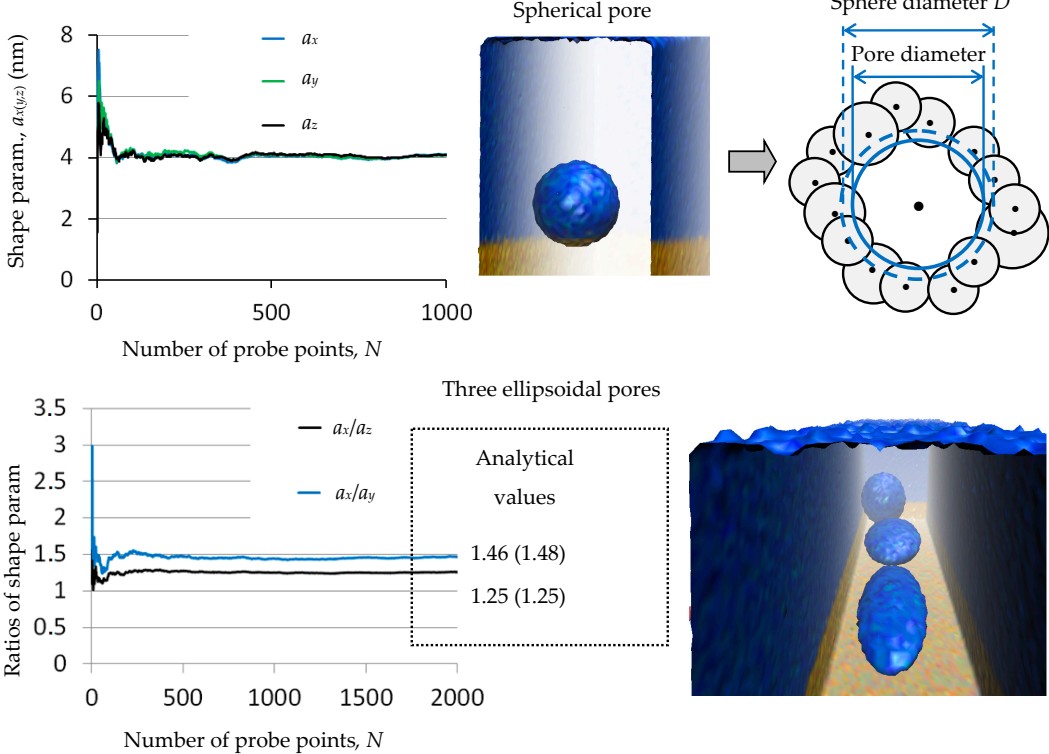

**Figure 5.** Result of the shape parameters calculations for the model problems. The convergence of the calculated values with an increase in the number of probe points is shown. The analytical and simulation values of shape parameters ratios coincide. The shape of the pores is shown on the right part of the figure.

The diameter of the spherical pore $D$ is taken equal to 6 nm. For a continuous medium, the integral on the right-hand side of Equation (10) can be calculated analytically. This gives $a_x = a_y = a_z = 3D/4 = 4.5$ nm. The results of numerical simulation are shown in the plots in the upper part of Figure 5. Values of $a_{x(y,z)}$ converge to approximately 4 nm with an increase in the number of probe points. The small difference between the results of analytical and numerical solutions is due to the difference between the pore surface and the spherical surface. Indeed, the pore surface is formed by the fragments of spheres centered on the atoms closest to the center of the pore (Figure 5, right part). Since these fragments are partially located inside a sphere of diameter $D$, the calculated shape parameters are smaller than the analytical parameters for a continuous medium.

The shape parameters are also calculated for the three ellipsoidal pores. The principal axes of all ellipsoids are directed along the coordinate system axes (Figure 3), the value of β is equal to 30°. Ellipsoids are placed inside a cluster so as to avoid their overlapping. To diverse the shape of the ellipsoids, values of their semi-axes are taken equal to $a_{x1;2;3} = 8; 7; 4$ nm, $a_{y1;2;3} = 4; 5; 5$ nm, $a_{z1;2;3} = 2; 3; 6$ nm.

The dependences of shape parameters ratios on the number of probe points are shown in the lower part of Figure 5. These ratios converge fast to the theoretical values with an increase in the number of probe points. Some differences between the numerical and analytical results are explained in the same way as for the spherical pore.

To apply the developed approach to the silicon dioxide films, the values of the atomic radii of silicon and oxygen atoms should be determined. As follows from Equation (5), the components of the dielectric constant and refractive index tensors depend on the free volume fraction of the film. The free volume is defined as volume outside spheres centered on the atoms of the cluster. Therefore, the free volume is fully determined by the Cartesian coordinates of the atoms and values of the atomic radii.

On the other hand, the dependence of free volume fraction on the deposition angle is determined by Equation (6). Thus, the value of the atomic radii should be chosen so as to reproduce this dependence.

Earlier in the porosity calculation [34] the van der Waals radii of oxygen and silicon atoms were taken equal to $R_{vdW}(O) = 0.152$ nm [35] and $R_{vdW}(Si) = 0.21$ nm [36]. However, these radii leads to $f_1(\alpha)$ values significantly higher than those obtained using Equation (6), see the bottom row in Table 1. This is due to the large number of small pores appearing between adjacent atoms if atomic spheres do not cover the entire volume of the film. Thus, the values of the atomic radii should be increased in comparison with the values of van der Waals radii. It was found that radii equal to 0.21 nm for oxygen and 0.27 nm silicon allow one to reproduce $f_1(\alpha)$ dependence obtained using Equation (6) (third and fourth rows, Table 1).

**Table 1.** Dependencies of the film density $\rho$ (g/cm$^3$) and free volume fraction $f_1$ on the deposition angle $\alpha$.

| A | 0° | 40° | 50° | 60° | 70° | 80° |
|---|---|---|---|---|---|---|
| $\rho(\alpha)$ | 2.40 | 2.35 | 2.24 | 1.92 | 1.75 | 1.36 |
| $f_1(\alpha)$ Equation (6) | 0 | 0.020 | 0.067 | 0.200 | 0.271 | 0.433 |
| $f_1(\alpha)$ [1] | 0.023 | 0.032 | 0.080 | 0.195 | 0.269 | 0.436 |
| $f_1(\alpha)$ [2] | 0.205 | 0.208 | 0.261 | 0.366 | 0.429 | 0.640 |

[1] Atomic radii, $R(O) = 0.21$ nm, $R(Si) = 0.27$ nm, [2] Van der Waals radii, $R_{vdW}(O) = 0.152$ nm [35], $R_{vdW}(Si) = 0.21$ nm [36].

The convergence of the pore shapes parameters in the thin film clusters with an increase in the number of probe points N is shown in Figure 6. A value of $N$ is approximately ten thousand, which is enough to calculate the values of $a_{x(y,z)}$. This calculation takes about ten minutes for clusters consisting of $10^6$ atoms. Since the algorithm has linear scaling with the number of atoms, it takes about one hour for clusters with $10^7$ atoms. As can be seen in Figure 6, an approximate estimation of ratios of $a_{x(y,z)}$ values can be done much faster. In any case the calculation of the of $a_{x(y,z)}$ values requires much less time than the atomistic simulation of the deposition process using MD or kMC methods. Besides, the algorithm can be easily parallelized since the probe points in different parts of the clusters can be chosen independently.

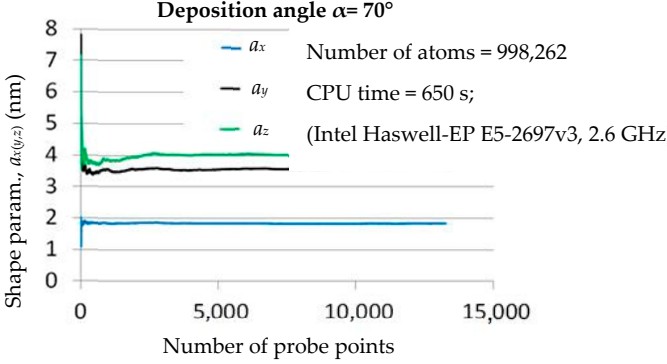

**Figure 6.** Dependence of the shape parameters on the number of probe points in Monte Carlo simulation.

Dependencies of the shape parameters and depolarization factors on the orientation of the coordinate system axes are shown in Figure 7. Due to the symmetry of the deposited structure, the differences in these values achieve maximum when the Z axis is directed almost parallel to the slanted columns (see the right part of Figure 7). This orientation of axes ensures the maximum value of $a_z$ close to 4 nm (see the left part of Figure 7). On the contrary, with this axes' orientation, the value of $a_x$ reaches its minimum, since the X axis is directed almost perpendicular to the elongated pores. Value of $a_y$ changes insignificantly with an increase of rotation angle β.

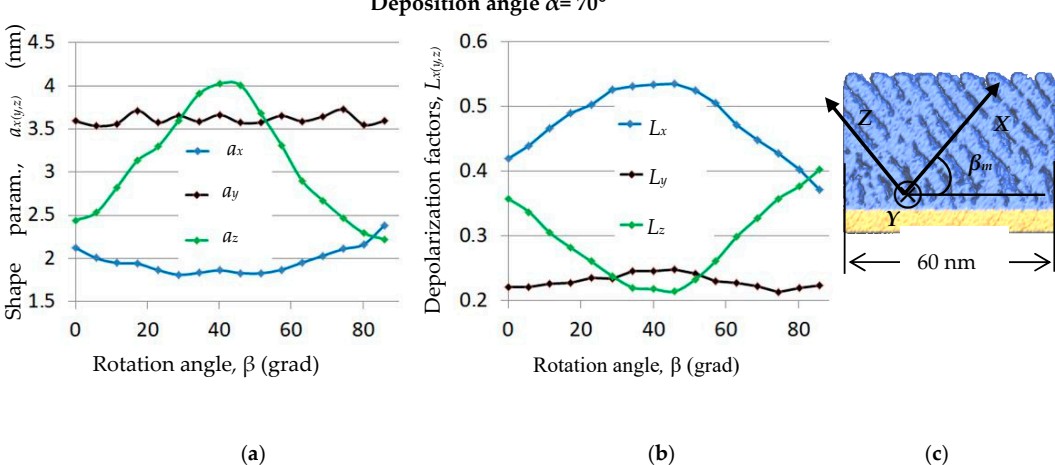

**Figure 7.** Dependencies of the shape parameters and depolarization factors on the rotation angle β, $β_m$ is the rotation angle at which the maximum value of $L_x$–$L_z$ is reached. (**a**) Values of shape parameters; (**b**) Values of the depolarization factors; (**c**) Orientation of the coordinate axes

The dependencies of the depolarization factors on the angle β are explained in the same way. In accordance with the theoretical results [9,16], the sum of all factors is constant and equal to 1. At a rotation angle $β_m$, the factor $L_x$ is maximum, since the ratio of $a_x/a_z$ is minimal, see Equation (3). The column tilt angle γ can be approximately calculated as 90° − $βm$ (see the right part of Figure 7), which gives about 50° for γ. The tangential rule [17] predicts γ = 53° for the deposition angle α = 70°, which is close to our value of 50°.

The results of the refractive index calculation are shown in Figure 8 and in Table 2 and are compared with experimental data known from the literature. For α = 50° the Δ$n$ value is less than 0.003, i.e., is at the level of anisotropy of the normally deposited $SiO_2$ films [32]. The dependences of the refractive index on the rotation angle β for the deposition angles α = 60°, 70°, 80° are similar. The difference Δ$n = n_z − n_x$ reaches its maximum at a certain value of the rotation angle β = $β_m$. The value of $β_m$ increases with increasing of α. This is explained by an increase of the tilt angle of the separated columns in GLAD films with the increase in the deposition angle [3]. Since the pores are elongated along these columns, the angle between the substrate plane and $Z$ semi-axis of the ellipsoids also increases.

As expected, the Δ$n$ value monotonically increases with increasing the deposition angle, due to the increase in the film porosity. The difference between the $n_z$ and $n_x$ values reaches a maximum value of 0.035 when α is equal to 80°, and $β_m$ is approximately 55°.

The calculated values of Δ$n$ are less than experimental ones for all of the deposition angles (Table 2). This can be explained as follows. As it is mentioned in Section 2.2, dimensions of the open pores in the vertical direction are limited by the upper boundary of the film. This leads to the underestimation of the $a_z/a_x$ ratio calculated for the coordinate system orientation corresponding to the rotation angle $β_m$ (Figure 7). To take into account, the contribution of the large elongated pores to the $a_z/a_x$ ratio and Δ$n$, the simulation with clusters of larger dimensions should be performed.

The calculated values of the refractive index $n = (n_x + n_y + n_z)/3$ are slightly higher than the experimental ones (Table 2). A possible reason for this is an underestimation of the porosity of the clusters deposited in our MD simulation runs. This also leads to the underestimation of the calculated Δ$n$ values compared to the experimental data, see the paragraph above.

The error in the refractive index in the experiments cited in Table 2 is about 0.005 [6]. When modeling, various sets of random numbers were used in the Monte Carlo method. This led to a difference in the refractive index values less than 0.002. The convergence of the calculated values with an increase in the number of probe points was ensured in our simulation experiments, see plots in Figure 6.

To summarize, the reached level of the simulation accuracy is close to that reported in Reference [14]. To increase the accuracy of the developed approach, it is necessary to increase the dimensions of simulation clusters. This requires the use of more high-performance computational recourses. It should be emphasized that the model does not have any adjustable parameters used to reproduce the experimental results.

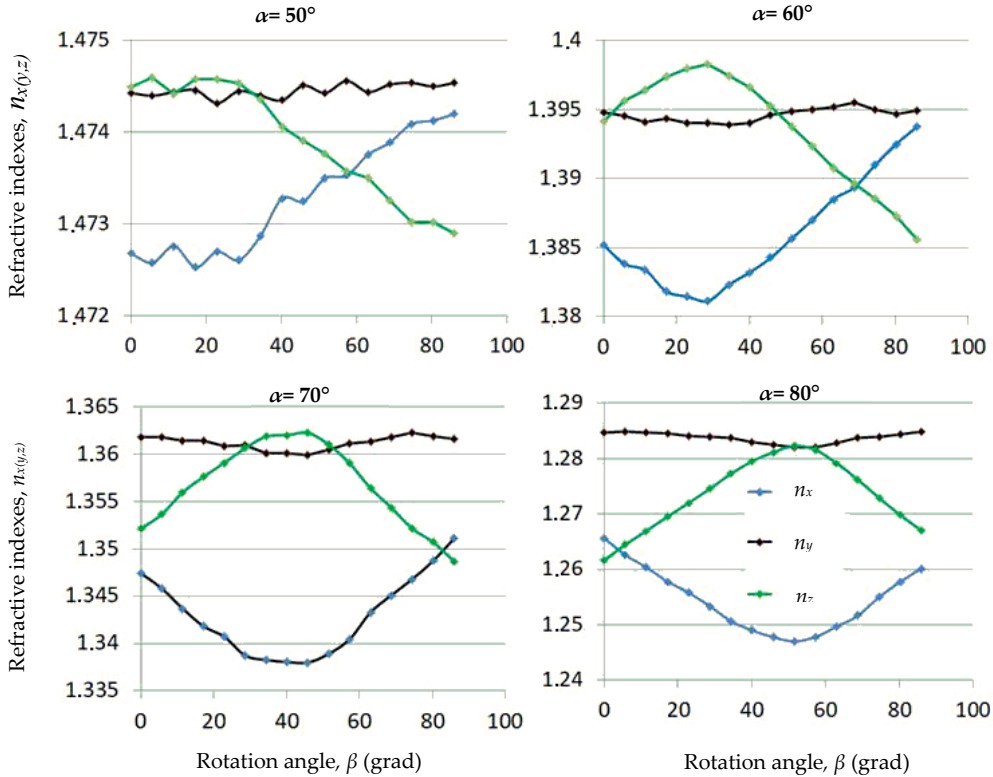

**Figure 8.** Dependences of the difference of the refractive index main components on the rotation angle for different values of the deposition angle $\alpha$.

**Table 2.** Dependencies of the difference between the components of the refractive index $\Delta n$ and refractive index $n$ on the deposition angle $\alpha$. The substrate matter, deposition method, wavelength at which the refractive index is determined, and substrate temperature are given in the notes below the table.

| A | 60° | 70° | 80° |
|---|---|---|---|
| $\Delta n$ (calc.) | 0.015 | 0.025 | 0.035 |
| $\Delta n$ (exp.) | 0.025 [1] | 0.04 [1] | 0.05 [1] |
| $n$ (calc). | 1.391 | 1.353 | 1.270 |
| $n$ (exp). | 1.42 [1]<br>1.31 [3]<br>1.33 [4] | 1.34 [1]<br>1.325 [2]<br>1.21 [3] | 1.22 [1]<br>1.20 [2]<br>1.16 [3]<br>1.25 [4] |

[1] [6]; indium tin oxide (ITO) coated glass substrate; electron beam evaporation; 532 nm; 330 K. [2] [37] fused silica substrate; 532 nm. [3] [38] fused silica substrate; electron beam evaporation; room temperature, 532 nm. [4] [39] *p*-AlGaN substrate; electron beam evaporation; room temperature, 365 nm.

## 4. Conclusions

In this study, the combined modeling of the optical anisotropy of porous silicon dioxide thin films was carried out. Atomistic clusters representing the structure of the films are obtained using the full-atomistic MD simulation of the deposition process. The pores in films structure are approximated by a set of ellipsoids having the same shape parameters and the same orientation. These parameters are calculated in the framework of the Monte Carlo based method, and are further used in the anisotropic Bruggeman effective medium theory to determine the difference of the main components of the refractive index $\Delta n$.

The $\Delta n$ values are calculated for the high-energy deposited clusters. Deposition angles vary from $0°$ to $80°$, which correspond to the normal and glancing angle deposition, respectively. Normally deposited film is considered as a homogeneous structure without void volume. The increase in the deposition angle leads to the decrease of the film density, due to pore formation. The maximum value of the difference of the refractive index main components is equal to 0.035. While the reached level of the simulation accuracy is close to that reported earlier [14], the calculated values of $\Delta n$ are still somewhat lower than the experimental ones. Possible reasons for this are discussed, and it is pointed out that even larger atomistic clusters should be considered to increase the accuracy of simulation experiments.

The developed method of combined modeling of the anisotropic properties of atomistic clusters can be applied to other oxide films. For the calculation of $\Delta n$ values, information is required about the deposition conditions—including the energy of the incoming atoms, the value of the deposition angle, and the substrate temperature.

**Author Contributions:** Conceptualization, A.V.T., F.V.G., V.B.S.; methodology, F.V.G.; software, F.V.G. and V.B.S.; validation, F.V.G.; formal analysis, F.V.G.; investigation, F.V.G.; resources, A.V.T. and V.B.S.; data curation, A.V.T. and V.B.S.; writing—original draft preparation, F.V.G.; writing—review and editing, A.V.T. and V.B.S.; visualization, F.V.G.; supervision, A.V.T. and V.B.S.; project administration, V.B.S.; funding acquisition, A.V.T. and V.B.S. All authors have read and agreed to the published version of the manuscript.

**Funding:** The work was supported by the Russian Science Foundation (Grant No. 19-11-00053).

**Conflicts of Interest:** The authors declare no conflict of interest.

## Appendix A

Consider a medium with $N$ ellipsoidal pores that have different shape parameters and are oriented in the same direction. For this case, the semi-axis value averaged over the pores volume is calculated as follows:

$$\langle a \rangle = \frac{\sum_{i=1}^{N} V_i a_i}{\sum_{i=1}^{N} V_i} \tag{A1}$$

where $V_i = (4\pi/3)a_i b_i c_i$ is the volume of the $i$-th ellipsoid, $a_i$, $b_i$ $c_i$ are the semi-axes of $i$-th ellipsoid.

On the other hand, the averaged pore dimensions parameter $a_x$ is calculated as follows (see Equation (10)):

$$a_x = lim_{N_{in} \to \infty} \frac{\sum_{i=1}^{N_{in}} a_{xi}}{N_{in}} = \frac{\sum_{i=1}^{N} \iint a_{xi}^2(y,z)dydz}{\sum_{i=1}^{N} V_i} \tag{A2}$$

where integration in each term is performed over the volume of the $i$-th pore. The boundary of the $i$-th ellipsoid is defined by the equation:

$$\left(\frac{x}{a_i}\right)^2 + \left(\frac{y}{b_i}\right)^2 + \left(\frac{z}{c_i}\right)^2 = 1 \tag{A3}$$

The $a_{xi}$ value is determined using Equation (A3):

$$a_{xi}(y,z) = 2a_i \sqrt{1 - \left(\frac{y}{b_i}\right)^2 - \left(\frac{z}{c_i}\right)^2} \tag{A4}$$

So:

$$\iint a_{xi}^2(y,z)dydz = \int_{-c_i}^{c_i}\left(\int_{-yi}^{yi} a_{xi}^2(y,z)dy\right)dz = \frac{3a_i}{2}V_i \tag{A5}$$

where $yi = b_i(1 - (z/c_i)^2)^{1/2}$. Substituting Equation (A5) into Equation (A2), we obtain:

$$a_x = \frac{3\sum_{i=1}^{N}V_i a_i}{2\sum_{i=1}^{N}V_i} = \frac{3}{2}\langle a\rangle \tag{A6}$$

The $a_{y(z)}$ values can be calculated in the same way. Thus, the ratios of the shape parameters are equal to the ratios of the averaged principal semi-axes:

$$\frac{a_x}{a_y} = \frac{\langle a\rangle}{\langle b\rangle}; \quad \frac{a_x}{a_z} = \frac{\langle a\rangle}{\langle c\rangle} \tag{A7}$$

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
