# Peer review of "Combined Modeling of the Optical Anisotropy of Porous Thin Films"

_coatings, doi:10.3390/coatings10060517_

Round 1
Reviewer 1 Report
In the present paper, a combination of methods is used to model the optical anisotropy of the glancing angle deposited thin films from SiO2. The authors use an earlier developed molecular dynamics procedure to model the growth of clusters in the initial stage of thin layer deposition. The next step by Monte Carlo simulation and the applying the effective medium theory are modeled pore sizes and shape, and the effective refractive index of the thin films.
I believe that the article presents new results and can be published in the Coatings journal after clarifying some points.
- Do the authors take into account in their model that the experimental results show that the slope of the film's columns and the angle of the incidence of vapors do not coincide and usually follow the so-called tangential or cosine rules.
- Despite the existing diverse of the existed experimental data in literature depending on the methods of deposition, substrate and etc., the obtained theoretical results are in the agreement with them. Authors may need to comment on the accuracy of the anisotropy of the refractive index they cite in Table 2 or how small deviations affect the initial values of their modeling of the final result.
- The conclusion must give answer of readers whether the proposed procedure is applicable in the event of a change in the conditions of postponement or in the case of other materials and what information is needed. In this case, the procedure would make it possible to predict in advance the optical anisotropy of the deposited thin layer.
- What is the algorithm the authors used for minimization of the potential in Eg.1.
I noticed some technical errors in the text listed below that need to be corrected.
1. What id definition of the angle β. Is it the same as βm in Fig.6.
2. The equation numbers in the following sentence (Page 5, row 162): “The values of Lx(y,z) and nx(y,z) are calculated using Eqs. (1,2). “ is not correct. Eq.1 gives interatomic interaction.
3. Figures must be improved. Some axes is not specified, legends and some symbols are
- Axes of Figure 5b are not clear. The legend in Figure 5b has to be given. The caption of the figure does not contain description of the figure.
- Symbol “X” is over cropped in scheme of Figure 6.
- The subscripts in the legend of Figure 7 are cut.
4. The text in paragraph Page 7 rows 226-232 is duplicated in the next paragraph Page 7 rows 233-239.
Reviewer 2 Report
This manuscript describes a modeling approach that can be used to investigate the optical anisotropy behavior of porous thin films, a particularly difficult thin film material type to explore both computationally and experimentally. In this report, the authors use both atomistic (molecular dynamics simulations) and continuum approaches, and calculates the refractive index using the anisotropic Bruggeman effective medium theory. The simulations are then compared with experimental data for validation and show good agreement.
Comments/Questions
- Line 44: "Up today". I'm not sure what this means. Could this be rephrased?
- Line 49: "Main components" should be "The main components . . ."
- Figure 1 caption: could you please provide more information in the caption. What is the main message? What is notable about the differences between the two angles of deposition (how does it change the structure)?
- Line 100: "Simulation is" should be "The simulation was."
- Figure 2 caption: please provide more details, restate the shape parameters and their definitions, etc. It would be helpful to the reader if the figure and caption could be understood on their own without reference to the main text.
- Equation 2: please define all variables in the equation.
- Figure 3: the flowchart of how the shape parameters are calculated is good in terms of the visuals, but the caption needs more detail. See comment 5, above.
- Figure 4: see comment 5, above.
- Figure 5: see comment 5, above. What is the main message from this? What is the main point of the figure?
- The paragraph in lines 226-232 is repeated.
- It does not appear that the simulations were experimentally validated using new experimental data? If so, the film growth and characterization methods need to be described in the manuscript, and the figures associated with the SiO2 films should be included (SEM of film surface, film thickness, ellipsometric measurements, etc.). It looks like values of n were found from the literature instead? I suppose this is a different approach than I expected when I read the abstract - when it said the model was experimentally validated, I expected the manuscript to include experimental work - film deposition and characterization.
- The authors self-cite 5 times out of 37 references. They seem to be appropriate citations describing their previous methodology.
